# The Fragility of Restoring Full Ordination for Tibetan *Tsunmas* (Nuns)

Darcie M. Price-Wallace 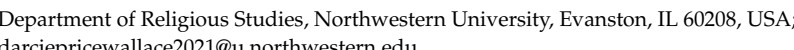

Department of Religious Studies, Northwestern University, Evanston, IL 60208, USA;
darciepricewallace2021@u.northwestern.edu

**Abstract:** Drawing from interviews with *tsunma*s (Tib. *btsun ma*, nun) living and practicing in Geluk, Kagyu, and Sakya institutions in Bihar, Himachal Pradesh, Ladakh, and Uttarakhand, India, between March 2017 and February 2019, this case study foregrounds *tsunma*s' heterogenous insights into the most ideal and acceptable ways for restoring *gelongma* (Tib. *dge slong ma*, Skt. *bhikṣuṇī*, fully ordained nun) vows. I argue that fragility, the quality of being breakable, underlies the history of *gelongma* vows in Tibet. Fragility, however, can also be generative. In this regard, fragility also signifies the possibility of restoring *gelongma* ordination for some *tsunma*s who are interested in receiving *gelongma* vows in India. This article examines Tibetan and Himalayan *tsunma*s' perspectives on the possible ways of restoring *gelongma* ordination for women in the Tibetan Buddhist tradition. Since the lineage of *gelongma*s ceased in Tibet, some *tsunma*s see this fragility as prohibiting restoration of *gelongma* ordination unless there is a way to re-establish these vows through the *Mūlasarvāstivāda Vinaya*, the monastic code adopted in Tibet, whereas other *tsunma*s perceive this fragility as an opportunity for other possibilities for *gelongma* vow restoration through innovative ritual practices such as dual-*Vinaya* ordination.

**Keywords:** Buddhist nuns; full ordination; *Vinaya*; gender asymmetry

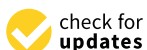



## 1. Introduction: Historical Background on the Topic of Full Ordination for *Tsunma*s in the Tibetan Buddhist Tradition

> *When the recognition of impermanence shakes us into accepting the certain demise of our body, then we really aspire to make the most of our life. The truth of impermanence becomes the wind at our backs, urging us not to squander the precious opportunity we have right now.* (Mingyur Rinpoche 2014, p. 112)

The question of full ordination for Buddhist women spans 2600 years beginning with the Buddha's foster mother Mahāprajāpatī's request for full ordination, a fragile issue even in the earliest canonical narratives. For example, in the chapter on Mahāprajāpatī within the Tibetan *Mūlasarvāstivāda Vinaya*, one of the Buddha's metaphors for initially rejecting her request to ordain is due to the presumed precariousness that women's ordination may bring upon the tradition:

> Ānanda, it is as follows. For example, a household in which there are many women and few men is easily attacked and overwhelmed by robbers and kidnappers. Likewise, Ānanda, if women go forth into the Dharmavinaya, it will not last long (Roloff 2020, p. 66).

In this canonical narrative, the act of women's ordination potentially renders the entire *Dharmavinaya*, the teachings of the Buddha, fragile, insofar as the Buddha's dispensation will not last quite as long as it would have otherwise. While the Buddha eventually granted Mahāprajāpatī's request, he did so by requiring these women to practice additional preventative measures, the eight *gurudharmas*, which similar to a dam, contains "women's faults" (Roloff 2020, p. 69). The women's acceptance of these eight *gurudharmas* negates the notion of the potentially shortened duration of the Buddha's dispensation although

these rules displace the women's *saṃgha* below the men's, establishing a very stark gendered asymmetry in Buddhist monasticism (Havnevik 1989, pp. 24–27; Gutschow 2004, pp. 169–74; Ngodrup 2007, pp. 152–53, 183–85; Anālayo 2010, pp. 81–82; Hüsken 2010, pp. 144–48; Salgado 2013, pp. 80–81; 2019, p. 5).[1]

According to Shayne Clarke, there are six extant *Vinaya*s belonging to different schools (Clarke 2015, p. 60). Today, only three ordination lineages remain: (1) *Theravāda*, (2) *Dharmaguptaka*, and (3) *Mūlasarvāstivāda* (Clarke 2015, p. 60). Each reflects the gradual emergence and geographical dispersion of the schools where the *Theravāda/Pāli* school remains most prevalent in southeastern Asia including Cambodia, Laos, Myanmar, Sri Lanka, and Thailand, the *Dharmaguptaka* endures in China, Hong Kong, Korea, Vietnam, and Taiwan, and the *Mūlasarvāstivāda* continues in Bhutan, Mongolia, Nepal, and Tibet (Sujato 2010, p. 36).[2] The *Therāvada/Pāli* lineage of fully ordained women appears to have ceased around the end of the 10th century in Sri Lanka whereas the *Mūlasarvāstivāda* ordination lineages for women possibly never entered Tibet in the 8th century by a dual *saṃgha* ordination but were practiced through a single *saṃgha* (Wijayaratna 2001, pp. 147–48; Price-Wallace 2022, pp. 203–19). Social movements to find ways to restore these full vows for women practicing through these two different ordination lineages have been well-underway for nearly forty years, enabling women seeking full ordination the opportunity to receive these vows through the *Dharmaguptaka Vinaya*, a school considered distinctive from their own *Theravāda* or *Mūlasarvāstivāda* ordination lineages.[3] The differences between these *Vinaya* lineages have become paired with nuances in doctrinal and soteriological paths between the three vehicles of Buddhism, *Theravāda, Mahāyana*, and *Vajrayana*, even though this is more representative of geographic dispersion than of stark differences between the actual guidelines for monastics in the *Vinaya* code itself.

Full ordination for women matters because without these vows, the walls of institutional immobility and gender asymmetry for women within Buddhist monasticism remain. When women cannot receive their full monastic vows, this limits their access to studying the complete Buddhist canon in their tradition, minimizes educational opportunities, and diminishes the possibility to become teachers for other *tsunma*s and the *saṃgha* and/or hold leadership roles in ways they have rarely done previously (Gutschow 2004, pp. 168–97; Gyatso 2010, p. 17; Zangmo 2022; Schneider 2022).

Restarting the *Mūlasarvāstivāda* ordination became a topic of discussion in the 1990s in meetings held by the Central Tibetan Government's Department of Religion and Culture in Dharamsala, India, and during transnational discussions at Sakyadhita Conferences and the International Congress on Women's Role in the Sangha in Hamburg (hereafter: the Hamburg Conference) in 2007 (Tsomo 1988; Mrozik 2009; Mohr and Tsedroen 2010; Roloff 2020). The Hamburg Conference examined the question of legality, precedent, and social acceptance for giving the *gelongma* vows, but ultimately, no decision was firmly reached.

During the Hamburg Conference, while the Dalai Lama consistently offered his support for Tibetan *tsunma*s to receive their vows outside of the *Mūlasarvāstivāda Vinaya*, he noted that a majority of support from senior Tibetan *gelong*s was necessary for re-establishing *gelongma* vows via the *Mūlasarvāstivāda Vinaya* (the 14th Dalai Lama 2010, p. 277). The question remained—which way of re-establishing these vows would be most acceptable? Or, put differently, the least fragile. Why fragile? As the Dalai Lama noted,

> When it comes to re-establishing the Mūlasarvāstivāda bhikṣuṇī ordination, it is extremely important that we avoid a split in the sangha. We need a broad consensus within the Tibetan saṅgha as a whole, and we need to address not only bhikṣuṇī ordination but subsequent issues as well (the 14th Dalai Lama 2010, p. 277).

Feminist scholar and activist, Sara Ahmed says that "the word *fragility* derives from *fraction*. Something is broken. It is in pieces" (Ahmed 2017, p. 180). I read the same concern into the Dalai Lama's statement on two levels. First, the *gelongma* lineage is broken. Second, he is concerned about abating a split, a breakage amongst the *saṃgha*. The Dalai Lama,

living in exile in India, did not want any fracturing amongst the *saṃgha* and presumably the community of Tibetans living abroad.

I argue that the perpetual delay on re-establishing *gelongma* ordination has become a wall or what Ahmed would call, a brick wall, "the institutional walls; those hardenings of histories into barriers in the present" (Ahmed 2017, p. 135). While Ahmed's work speaks specifically to the limitations of diversity policies that aim to address institutional racism and sexism in Australian and British universities, her work on the institutional barriers that sustain immobility and prevent substantial change provides a way for thinking through the metaphorical fires and vases that underlie my interlocutors' concerns with *gelongma* ordination (Ahmed 2017, pp. 174–75). Delaying *gelongma* ordination is an institutional barrier that Tibetan monastic women come up against. It is a wall.

But what is the cost of shattering this wall? While the Dalai Lama is very supportive of finding pathways for *tsunma*s' education and ordination, he is also concerned, as are my interlocutors. The Dalai Lama suggests shattering this institutional wall may mean a split. To address the idea of shattering barriers, Ahmed segues into fragility. Fragility has many nuances that I detail throughout the article. Fragility is a useful interpretive construct for thinking about *gelongma* ordination since the breakage of this *Mūlasarvāstivāda* ordination lineage for women exhibits a "shattering of possibility." This means that its breakability is reflective of its fragility and "what breaks off is on the way to becoming something" (Ahmed 2017, pp. 168, 186). Thus, the question of restoring *gelongma* ordination points to breakability as well as how fragility points to the possibility of something else. Alongside the voices of my interlocutors, I pull in Ahmed's language as an interpretive tool. In addition, inspired by Ahmed, I make use of literary associations and their stories as a way to move between how "a breakage is often accompanied by a story, a story of what breaks when something breaks, or an explanation of what is behind a breakage" (Ahmed 2017, p. 165). In this regard, I take a non-traditional approach to thinking with and through my interlocutors' voices by drawing from a fictional narrative about a Japanese Buddhist nun, a Super Nun. My interlocutors told me stories about breakage, as does the fictional nun, Jiko, and her granddaughter Nao. These stories held alongside each other provide an interpretive lens for possibility in light of the fragility of *gelongma* ordination.

## 2. Re-Establishing the Tradition for Tibetan *Tsunma*s: Lighting a Fire with One Stick

The religious head of the Karma Kagyu school of Tibetan Buddhism, the 17th Gyalwang Karmapa, Ogyen Trinley Dorje (the 17th Karmapa, hereafter), intends to reinstate *gelongma* (Tib. *dge slong ma*, Skt. *bhikṣuṇī*, fully ordained nun) vows of Tibetan *tsunma*s (*btsun ma*, nun)[4] in his religious lineage in India (Roloff 2020; Karmapa 2022; Price-Wallace 2022).[5] This ordination requires several years to complete and is controversial for several reasons. Presently, *tsunma*s cannot receive their *gelongma* vows through the textual prescriptions of the *Mūlasarvāstivāda Vinaya*, the only monastic disciplinary code adopted in 8th century Tibet (Tsering 2010, p. 168). These ritual prescriptions require twelve *gelongma*s and ten *gelong*s (*dge slong*, *bhikṣu*) to confer the 364 vows[6] that guide the full extent of monastic women's ethical conduct and practice. Without a continuous *gelongma* lineage in Tibet, re-starting the lineage is like starting a fire with only one stick.

Conflicting opinions about restarting this metaphorical fire hinder consensus among stakeholders. The dominant narrative among conservative Tibetan monastic scholars demands the above prescriptions for the brightest, purest fire, essentially rendering it impossible to light in the absence of full ordination. These arguments hinge on the centrality of transmitting flawless, perfect vows without intermingling different lineages of *Vinaya* (Tsering 2002). Another contemporary monastic *Vinaya* scholar, Geshe Rinchen Ngodrup (2007, p. 9), foregrounds uncommon canonical precedents for bestowing *gelongma* vows by relying only on fully ordained monks for the ritual. This argument draws from exceptional situations in the *Vinaya* but has not garnered social acceptance since it is believed that the monks performing the ritual incur an infraction (Tsering 2010, pp. 168–71; Ngodrup 2007, p. 53).[7] The 70th Je Khenpo, Tulku Jigme Chhodea, of Bhutan, recently lit this metaphorical

*gelongma* fire with one stick. He performed a single *saṃgha* ordination when he gave *gelongma* ordination to more than one-hundred and forty *tsunma*s on 21–23 June 2022 (Zangmo 2022). In his final remarks after the ordination, he told the *tsunma*s that, "Each one of you have received the Bhikṣuṇī ordination need not have any doubt at all about whether you have obtained the vows or not" (Je Khenpo 2022, p. 11). His address speaks to the fact that giving the full ordination through the single *saṃgha* is not the creation of a new ritual practice but adheres to the exceptions in monastic code. As *Vinaya* scholar Clarke notes, because "Vinayas address a large range of exceptional situations," their redactors of *Vinaya*s had to deal both with ideals and deviations (Clarke 2010, p. 232).

Similarly, contemporary Buddhist leaders and *Vinaya* scholars have just as many variable opinions on differentiating between ideal and acceptable practices in the Buddhist community in India. For instance, contemporary *Vinaya* scholars such as Acharya Geshe Tashi Tsering and Geshe Dawa make canonical arguments that a *gelongma saṃgha* is necessary for the vows to be "flawless, perfect" (*nyes med phun tshogs*) according to tradition (Tsering 2010, pp. 175–79; Dawa 1999, pp. 15–17). The *gelongma saṃgha* gives the novice *dge tshul ma*, Skt. *śrāmaṇerikā*) and subsequently, the probationary precepts (*dge slob ma*, *śikṣāmāṇā*), which are observed for two years (Roloff 2020, pp. 194–97). At the time of the full vows, the *gelongma saṃgha* gives the celibacy vows (*tshangs spyod nyer gnas, brahmacāryopasthānasaṃvṛti*) on the day of the ordination as delineated in the *bhikṣuṇī* ordination procedures documented in the *Kṣudrakavastu*, which provides the stages for the bestowal of the precepts (Ngodrup 2007, p. 55; Tsering 2002, p. 169; 2010, p. 175; Tsedroen and Anālayo 2013, p. 762; Roloff 2020, pp. 83, 177–272).[8]

However, a more moderate position, such as the Karmapa's, posits a non-traditional approach using two different *Vinaya*s as a way to include both the women and men's *saṃgha*s. On 11 March 2017, the Karmapa initiated an ecumenical ritual with Taiwanese *gelongma*s using the *Dharmaguptaka Vinaya* to ordain Tibetan and Himalayan *tsunmas*, starting a potentially controversial fire with sticks from two different trees. At that time, Taiwanese fully ordained women administered the novice vows (*dge tshul ma*) to nineteen ordinands. There has been no movement or discussion about the subsequent stages for giving the *gelongma* vows via this dual-*Vinaya* approach, and the Karmapa recently mentioned its continuance during the 8th Arya Kshema in April 2022 when he noted he wished to move forward with this ordination ritual in the future (Karmapa 2022). Is this fragile fire the Karmapa started still burning or will the Je Khenpo's single *saṃgha* ordination provide precedent and support for future *gelongma* ordinations?

### 3. Interlude: A Tale for the Time Being

Nao Yasutani, one of the protagonists in Ruth Ozeki's novel, *A Tale for the Time Being*, journals her transformative events stemming from her experiences of sexual violence in high school, her father's attempted suicide, and meeting and staying with her great-grandmother, the nun Jiko. Nao writes:

> Okay, here's what I've decided. I don't mind the risk, because the risk makes it more interesting. And I don't think old Jiko will mind, either, because being a Buddhist, she really understands impermanence and that everything changes and nothing lasts forever. Old Jiko really isn't going to care if her life stories get written or lost, and maybe I've picked up a little of that laissez-faire attitude from her. When the time comes, I can just let it all go (Ozeki 2013, p. 27).

Nao references the risk of writing Jiko's life story, only to have it lost. Nao tries to embrace a similar attitude towards her own writing to mirror Jiko's embodiment of the Buddhist concept of impermanence. Nao, however, vacillates as to whether she "can just let it all go" while also commenting more broadly on sexual violence against women. Nao fears that her reader will toss her journal aside like one more victim. She writes, "What if you just think I'm a jerk and toss me into the garbage, like all those young girls I tell old Jiko about, who get killed by perverts and chopped up and thrown into dumpsters, just because they've made the mistake of dating the wrong guy?" (Ozeki 2013, pp. 26–27). She's

asking, what will you do with her words? What will you do with her and Jiko's stories? What will you do to address the stories of survivors of sexual violence?

Ozeki's novel still haunts me. Even so, I still tell everyone to read it especially because the ending perplexes me. Nao talks to her reader so that you feel like you are with her. I was with Nao. Even now I am with Nao. Nao would appreciate this play on words. She makes a similar joke somewhere in her journal. Nao speaks to me because she threads issues of male power, abuse, sexual violence, and silence on these actions throughout her journal. The pervasive realities of sexual violence fill this fictional sixteen-year-old Japanese girl's journal.

In the novel, I identify with Ruth, the other protagonist, who has found Nao's journal and shares it with the reader. Ruth, in stark contrast to Nao, receives support from men, such as her partner Oliver, who embraces pro-woman values. By pro-woman values, I mean attitudes that reflect a positive, supportive orientation towards women while acknowledging the limitations of sexed binaries and hierarchies pervasive among and within institutional structures more broadly (Langenberg 2018, pp. 1–24; Padma'tsho and Jacoby 2020, pp. 1–19; Price-Wallace 2022, pp. 416–19). Ruth reads Nao's journal to Oliver, who appears clueless about the sexual violence Nao experiences (Ozeki 2013, pp. 293–96).

As Ozeki alternates between Nao's journal and Ruth's reading of fragments of that journal, distinctive images emerge about how women confront sexual violence compared with the male characters who only slowly recognize the ways they failed to address perverse and very real sexually violent experiences. Nao's father says to her, "'I let you down,' he said. 'I was twisted up with my guilt. I wasn't there for you when you really needed me'" (Ozeki 2013, p. 388).

### 3.1. Objectives for the Time Being

This interlude into *A Tale for the Time Being* may seem tangential, but Ozeki's characters help me think about sexual violence, monastic women, and fragility in Buddhism. On the one hand, Ozeki's characters, like Nao's father, deal with sexual violence with further violence, such as an attempted suicide. This complicates the reader's own affective response regarding the sexual violence Nao experienced in her high school. Who does the reader feel more sympathetic towards? In her exploration of misogyny, philosopher Kate Manne notes, "Many people feel that men are entitled not just to be deemed innocent until proven guilty, but to be deemed innocent, period, regardless of their misdeeds" (Manne 2020, p. 12). Eventually, Ozeki's male characters do come around and acknowledge their own initial obliviousness towards the male hostility women and girls face, which polices and enforces gendered expectations (Manne 2020, p. 10; Tsomo 2019, pp. 304–5). Both Nao's father and Oliver eventually recognize how they let down the women around them by failing to acknowledge systems of abuse and power that harm women. In Ahmed's terms, these men both built and sustained a brick wall, an institutional barrier, that was harmful for all parties.

### 3.2. A Time for Gelongma Ordination

This article is actually about *tsunma*s, like Jiko, a character who knows how to live with fragility. I see parallels in how Jiko's character lives with fragility and how Ahmed explores fragility's nuances. Ahmed notes, "Fragility: the quality of being breakable. Fragility: when being breakable stops something from happening . . . . A break becomes the realization of a quality assumed to belong to something; breaking as the unfolding of being. And this is difficult: the assumption of fragility can make something fragile" (Ahmed 2017, p. 169).

Like Ahmed, I define fragility as the characteristic of being breakable (Ahmed 2017, pp. 168–69). Breakable implies impermanence, a quality of things not lasting, not durable—fragile. Everything is fragile—some things more or less than others (Ahmed 2017, p. 164). A central Buddhist teaching is that all conditioned things are impermanent, and contemplation on impermanence becomes an antidote to suffering. In the 19th century

Nyingma master and scholar Patrul Rinpoche's well-known "written guide" (Tib. *khrid yig*) on the oral explanation of the Buddhist teachings, he states,

> Whatever is born is impermanent and is bound to die.
>
> Whatever is stored up is impermanent and bound to run out.
>
> Whatever comes together is impermanent and is bound to come apart.
>
> Whatever is built is impermanent and bound to collapse.
>
> Whatever rises up is impermanent and bound to fall down.
>
> So, also, friendship and enmity, fortune and sorrow, good and evil, all the
>
> thoughts that run through your mind—everything is always changing (Patrul Rinpoche 1994, pp. 46–47).

The connection I want to make here is to think about how all conditioned things, like a *gelongma* ordination lineage, are impermanent and therefore fragile—breakable, ever changing. Yet, impermanence is not necessarily a generative positive characteristic in Buddhism; it may also be destructive, degenerative. As Buddhist teacher Chokyi Nyima Rinpoche writes, "The world is impermanent. One day everything we know will be gone. That's simply how it is. Everything ends and ceases to be. Deep down, we know this already; we just don't like to think about it. But in fact, everything changes from one moment to the next" (Chokyi Nyima Rinpoche 2018, p. 23). Chokyi Nyima Rinpoche also speaks to how recognizing impermanence is the foundation for Buddhist practice and therefore generative for a practitioner: "Understanding impermanence is the basis for all that is good, wholesome, joyful, and great. In that way, impermanence is our greatest teacher and our foremost source of inspiration" (Chokyi Nyima Rinpoche 2018, pp. 50, 82). Recognizing the impermanence of phenomena as key for anyone seeking to understand the Dharma suggests that the evolution of impermanence promises the prospect of transforming the practitioner's mind. I see similar parallels between the Buddhist understanding of impermanence and Ahmed's notion of fragility, which she positions as potentially generative. Singer–songwriter Leonard Cohen sings about the generative potential of broken things, "There is a crack, a crack in everything/That's how the light gets in/That's how the light gets in."[9]

I argue that restoring *gelongma* ordination vows via the *Mūlasarvāstivāda Vinaya* had remained stalled because of the fragility of its *gelongma* lineage, which depends upon (1) a continual *gelongma saṃgha* for giving vows with a *gelong saṃgha* or (2) solely on a *saṃgha* of *gelong*s.[10] Regarding re-establishing *gelongma* ordination, the first option is regarded as something that is broken, such as a lineage, that stops something from happening. Due to the absence of *gelongma*s in Tibet since Buddhism's introduction in the 8th century, the *gelongma* lineage never began in Tibet (the 14th Dalai Lama 1988, p. 44). The second option suggests how "a break becomes the realization of a quality assumed to belong to something" (Ahmed 2017, p. 168). By this, I mean that only relying on the *gelong saṃgha* illustrates how the broken *gelongma* lineage belongs to one *Vinaya* for Tibetan and Himalayan *tsunma*s; thus, one option for its continuance is reliance on what does exist, the *gelong saṃgha*.[11]

A third option entails a dual-*Vinaya* ceremony, which integrates fully ordained women monastics who hold precepts from the *Dharmaguptaka Vinaya* with *gelong*s ordained via the *Mūlasarvāstivāda Vinaya*.[12] Yet, this option implies "the assumption of fragility can make something fragile" because *Vinaya* scholars like Geshe Tashi Tsering question "whether the lineage of Chinese Bhikshunis initially ordained by a group of Sri Lankan Bhikshunis and Bhikshus that remains extant in China is pure and unbroken" (Tsering 2002, p. 171).[13] Since there are doubts among some contemporary *Vinaya* scholars about the viability and continuity of the lineage of fully ordained monastic women under the *Dharmaguptaka Vinaya*, it is also fragile. Therefore, any dual-*Vinaya* ordination by implication rests upon multiple fragilities.

There had been a prolonged silence on any future plans for the Karmapa to resume his initiative for restoring *gelongma* ordination, but he recently suggested he will continue

with the process once travel eases up amid the pandemic ([Karmapa 2022]).[14] Now, since the Je Khenpo of Bhutan gave the *gelongma* vows, is *gelongma* ordination still fragile? I argue, fragility, still underlies this topic of *gelongma* ordination because the history of *gelongma* ordination has become like a wall, an institutional barrier, and as Ahmed says, "histories that have become hard, histories that leave some more fragile than others" can become a thread for thinking through the things deemed breakable, like a *gelongma* lineage ([Ahmed 2017], p. 164). But what are *tsunmas'* perspectives?

Now, I turn to *tsunmas'* responses to questions about *gelongma* ordination to illustrate the fragility of *gelongma* ordination. This case study draws from select interviews with *tsunmas* living and practicing in Sakya, Kagyu, and Geluk institutions in Bihar, Himachal Pradesh, Ladakh, and Uttarakhand, India, between March 2017 and February 2019. First, through these *tsunmas'* voices, I explore the fragility of the *gelongma* lineage in Tibet. Second, I consider how contemporary *tsunmas'* responses to questions about full ordination illustrate the fragility of restoring *gelongma* ordination in the present. I depart from *tsunmas'* perspectives to think about the 17th Karmapa's perspective on the form and fragility of vows while pondering his erstwhile initiative to introduce *gelongma* ordination in the Tibetan tradition. By way of conclusion, I consider Ahmed's notion of fragility, "when being breakable stops something from happening" ([Ahmed 2017], p. 169). I conclude with and think of the fictional *tsunma*, Jiko, with Nao, and my interlocutor Geshema Jigme's advice for dealing with anything broken.

### 4. Fragility of a Tradition: The *Gelongma* Lineage in Tibetan History

In my case study, *tsunmas* living in India narrate the history of the *Mūlasarvāstivāda Vinaya* and ordination in Tibet and sometimes they present their own ideal for establishing a *gelongma* lineage as it relates to their knowledge of their community's needs and their understanding of *Vinaya*. Because *tsunmas* do not have *gelongma* ordination, they can only study parts of the *Vinaya* ([Schneider 2022]). Even though they are not authorized to study *Vinaya*, it still shapes their experience as holders of monastic discipline ([Blackburn 2001], p. 11; [Langenberg 2018], pp. 17–21). Thus, *tsunmas* still constitute what Anne Blackburn calls textual communities:

> Individuals who think of themselves to at least some degree as a collective, who understand the world and their appropriate place within it in terms significantly influenced by their encounter with a shared set of written texts or oral teachings based on written texts, and who grant special social status to literate interpreters of authoritative written texts ([Blackburn 2001], p. 12).

Blackburn takes up the term textual communities as she examines eighteenth-century Lankan Buddhism to illustrate how traditions are dependent on local communities ([Blackburn 2001], p. 10). Her work on textual communities helps us think about interpretive authority within textual communities and those who allow for interpretive shifts ([Blackburn 2001], p. 12). In the Tibetan context, lineage leaders, "going forth" or novice ordination preceptors, nunnery abbesses and abbots, philosophy teachers, and/or nunnery disciplinarians interpret texts to and for *tsunmas*. While *tsunmas* also study these texts themselves in classes and integrate their understanding through the practice of debate, they inherit interpretations, especially of *Vinaya*. Much like Blackburn's notion of textual communities in Sri Lanka, the *tsunmas* in this case study demonstrate that, "Although members of a given textual community are oriented by and toward shared texts, their interpretations of these texts are not homogenous" ([Blackburn 2001], p. 12).

Regarding history, the majority of *tsunmas* in this case study speak generally about the non-existence of *gelongmas* in Tibet because there is no evidence of a lineage of *gelongmas* ordained through *Mūlasarvāstivāda Vinaya* ([Price-Wallace 2017]).[15] Löpon Wangmo and Tsunma Tenzin's narratives speak to the fragility of the *gelongma* lineage as being breakable and therefore prohibiting something from happening for its restoration. These *tsunmas'* interest in *gelongma* ordination emerges from a wish for ordination solely through the *Mūlasarvāstivāda Vinaya*. Tsunma Sonam Khacho and Tsunma Yangchen perceive the

fragility of the *gelongma* lineage in line with Ahmed's other definition of fragility—"breaking as the unfolding of being," meaning that due to the break, they see other possibilities such as dual-*Vinaya* ordination.

*4.1. Single Vinaya: Significance of the Mūlasarvāstivāda Vinaya*

4.1.1. Löpon Wangmo

Löpon Wangmo, the principal of the Sakya nunnery in Uttarkhand, toured me around the campus where nearly sixty *tsunmas* reside for advanced study. Struck by the size of the well-landscaped campus, I marveled at the lush, subtropical foliage underneath the canopy of trees encircling the quadrangle, whose main focal point drew my eye to the main temple. Inside the temple, the golden statue of Shakyamuni Buddha overlooked rows of carpeted benches and reverently awaited the commencement of evening prayers. The ornate red and maroon columns defined the open spaces, and blue ceilings decorated with dharma wheels contrasted with the green-painted walls, off-setting the meticulously detailed images of buddhas, dharma protectors, lineage leaders, sacred sites, and other sources of refuge and religious inspiration. Compared with the other nunneries I had visited in India, this was the largest, most ornate temple set amid a manicured and well cared-for campus.

An hour into our formal interview, I asked Löpon Wangmo, "Would you ever be interested in full ordination?" Löpon Wangmo replied, "Yes, I am. Your question is a direct yes or no question." We both smiled and she replied, "My answer is also direct, 'yes.'"[16] Later in our interview, Löpon Wangmo clarified, "Generally, this *gelongma* ordination is not about interest but about history." She proceeded to tell me the history she inherited, the common narrative among her textual community about the *Mūlasarvāstivāda Vinaya* as it relates to ordination in Tibet.

I asked, "Why might *tsunmas* want to only ordain under the *Mūlasarvāstivāda Vinaya* and not have an ecumenical ordination that includes fully ordained monastic women under the *Dharmaguptaka Vinaya*?" Löpon Wangmo explained:

> The Tibetan King, Trisong Detsen (*khri srong lde btsan*), invited Khenchen Śāntarakṣita (*mkhan chen zhi ba 'thso*) from Nalandā University in India. Śāntarakṣita believed the *Mūlasarvāstivāda Vinaya* would be best for Tibetans. He ordained seven monks, and this has been the practice. Later, when Atiśa came to Tibet, he was asked to give ordinations. Since Śāntarakṣita had introduced the *Mūlasarvāstivāda* lineage, Atiśa thought passing a lineage from another tradition would be inconsistent. So, he did not give an ordination because having two different lineages in one country would be confusing. Importing a new tradition would not be a big problem if it is a faultless ordination. Nonetheless, since in Tibet we only have had one tradition from the beginning, I think Tibetan *tsunmas* would love to have ordination through the *Mūlasarvāstivāda Vinaya*; and they wish for a revival of that *gelongma* lineage. If practices are in one lineage, it would be nice. Sometimes all fruits are mixed and maybe we get a good juice, but if one tradition mixes with another tradition would be a funny flavor, particularly the ceremony chants![17]

Löpon Wangmo's response implies fragility in the sense that something being break-able stops something from happening. One, the *Mūlasarvāstivāda Vinaya* is the only lineage practiced in Tibet. As the Dalai Lama has stated, "Since the journey was very difficult in early times, *bhikṣuṇī*s [*gelongma*s] were not able to come to Tibet" (the 14th Dalai Lama 1988, p. 44). Thus, the lineage broke when *gelongma*s did not travel to Tibet and this stopped the lineage from continuing. Two, Löpon Wangmo considers a second *Vinaya* tradition as potentially problematic due to the mixing of rituals. Where Geshe Tashi Tsering doubted the continuity of the *Dharmaguptaka Vinaya* lineage for women, Löpon Wangmo accounts for differences in ritual practice. Three, she assumes that *tsunmas* would only want *gelongma* vows from the *Vinaya* most commonly observed among Tibetan monastics. Lastly, she used the Tibetan word for faultless or flawless (*nyes med*) when speaking about other *Vinaya* traditions. Here, this mirrors Ahmed's suggestion that "the assumption of fragility

can make something fragile." Löpon Wangmo presents concern that another tradition's ordination could be with fault, fragile to a degree. Relying on another tradition would thus make restoring vows doubly fragile. According to Löpon Wangmo, these possible paths to restoring *gelongma* ordination render these vows fragile.

### 4.1.2. Tsunma Tenzin

Tsunma Tenzin, a forty-year-old Geluk *tsunma* from Zangskar, and I met while she attended the Dalai Lama's teachings in Bodh Gaya. Her main responsibilities in her nunnery include teaching the younger *tsunmas.* When I asked about a *gelongma* lineage in Tibet, Tsunma Tenzin said, "There have never been any *gelongma*s in Tibet. Now, we have many *geshemas*, [an advanced degree for *tsunmas* first conferred in 2016], but we do not have *gelongma* ordination. His Holiness [the Dalai Lama] says that he cannot do anything about *gelongma* ordination since it has to do with monastic rules as taught by the Buddha. The *geshema* degree, however, is an academic achievement based upon a curriculum and exams."[18] On the one hand, Tsunma Tenzin speaks about fragility of the *gelongma* institution. On the other hand, recently conferred *geshema* degrees highlight evolving standards of *tsunmas'* education in India.

While she speaks of the fragility of the *gelongma* lineage in Tibet, like Löpon Wangmo, Tsunma Tenzin did not see a dual-Vinaya ordination as viable. She did not reference differing ritual practices but instead emphasized fragile Sino-Tibetan political relations. She said, "We could not take the *Dharmaguptaka Vinaya* vows mainly because of our nationality. We are Tibetans. I cannot see any other reason. In Tibet, we only have the *Mūlasarvāstivāda Vinaya* tradition and the Chinese have the *Dharmaguptaka Vinaya*. Although all the eighteen *nikayas* are Buddhist, but because of politics, the Chinese seem to be our rivals."[19] Later, this led into Tsunma Tenzin and I discussing the Karmapa's plans for a dual-*Vinaya* ordination. She, however, concluded that "since we do not have the twelve *gelongma*s, we cannot do the ordination under the *Mūlasarvāstivāda Vinaya*."[20] Thus, based on its fragility—its history of brokenness—the *gelongma* ordination cannot be restored, according to both Löpon Wangmo or Tsunma Tenzin.[21] As a point of contrast, Tsunmas Sonam Khacho and Yangchen offer a different perspective on the fragility of *gelongma* ordination and how its breaking could be an unfolding of new possibilities through a dual-*Vinaya* ordination.

### 4.2. Dual-Vinaya: Mūlasarvāstivāda and Dharmaguptaka Vinayas

### 4.2.1. Tsunma Sonam Khacho

Tsunma Sonam Khacho's Kagyu nunnery sat atop a very windy road amid the coniferous and oak-tree filled hills where shades of green soften the rocky cliffs dappling northern Himachal Pradesh. Tsunma Sonam Khacho and I had met several times as I frequently visited her nunnery. It was during our third meeting that Tsuma Sonam Khacho, a *gelongma* who received her full vows in Hong Kong in the 1980s, described the challenges her parents encountered when they all came from Tibet to India. She had joined Tsunma Pema's interview and sat listening and sharing her own perspectives. We had not talked much about Tsunma Sonam Khacho's early life, but she said, "no one knew how to communicate with the locals" and "we only had dirty water." [22] She was really young at the time she left Tibet, but vividly remembered, "the water was cleaner in Tibet since it came from the mountains, and we were nomads. The water came from pipes in India."[23]

Tsunma Sonam Khacho did not share many details beyond this narrative about her pre-ordination life, but she did offer her insights into Tibetan history as it relates to restoring *gelongma* ordination at present:

> The *gelongma* lineage declined in Tibet. So, the contemporary issue about *gelongma* ordination raises questions for many people since it is a new and strange phenomenon. If *gelongma*s like us practice according to tradition, gradually people will not complain and the *gelongma* ordination will flourish. On the contrary, if we

do not practice well, negative attitudes or gossip will spread in the community. That is not a good thing for anyone because it is all negative karma.[24]

Her last sentence is a reference to the ten actions to be avoided such as worthless chatter since it risks causing problems among individuals and community members (Patrul Rinpoche 1994, pp. 108–9). She highlights the fragility of a *gelongma* lineage in a new light—its possibilities. Rather than fragility preventing something from happening, this full ordination did happen for her and seven other *tsunmas* who received their vows via the *Dharmaguptaka Vinaya* in the 1980s (Havnevik 1989; Price-Wallace 2022).[25] She also spoke in terms of the Dalai Lama and Karmapa's efforts for restoring the *gelongma* lineage. She said, "Individuals like His Holiness [the Dalai Lama] and Gyalwang Karmapa are not just Buddhas, they are omniscient Buddhas who can understand the situations of all three times. Therefore, in my view, they must have had a vision that it is essential to practice *gelongma* vows now for the future of the Buddhist tradition."[26] For Tsunma Sonam Khacho, whose own *gelongma* ordination remains uncommon among Tibetan and Himalayan *tsunmas* in India, this statement not only signals to what she sees as important for the future of Buddhism, but also speaks to how monastic communities regard the authoritative figures heading lineages—as Buddhas. For this reason, the Karmapa's plans for an ecumenical dual-*Vinaya* ceremony coalesces with how she understands the fragility of the tradition. If new possibilities do not unfold from the breakage of the *gelongma* ordination lineage in Tibet, what will become of the Buddhist tradition?

However, she also points to another issue, the fragility of one's own vows and practice. If she and the other *gelongma*s do not practice in an exemplary manner, then the fragility of their own vows impacts the fragility of the tradition as well. Tsunma Sonam Khacho was not the only *tsunma* to express concern about the potential fragility of one's own vows; Tsunmas Yangchen and Pema also express this as well.

### 4.2.2. Tsunma Yangchen

Tsunma Yangchen, a Kagyu *tsunma* from Kham, had lived in India for twenty years. Her nunnery in the Uttarkhand valley felt like a small, gated community opening up amid surrounding pastureland. When I asked her about receiving *gelongma* ordination, she stated, "First I must focus on observing the vows I have and examine if I would be able to hold more vows well. Once I am confident enough and ready for *gelongma* ordination, I would like to receive it."[27] Her answer reflects her concerns about the fragility of vows rather than an emphasis on the fragility of the tradition. Later, however, Yangchen said, "Taking vows from another tradition [such as the *Dharmaguptaka Vinaya*] fills the void in our tradition; later we could also promote the vows within our own by the *Mūlasarvāstivāda Vinaya*. I think it is good, and we should try it."[28] Her approach echoes Ahmed's "unfolding of being." She sees the dual-*Vinaya* ordination as the key to transitioning to a future where a dual-*Vinaya gelongma saṃgha* who practice solely by the *Mūlasarvāstivāda Vinaya* could then ordain other *gelongma*s together with *gelong*s through the *Mūlasarvāstivāda Vinaya*. Tsunma Pema, in contrast, spoke with me not about *gelongma* vows, but about the vitality of one's own vows on the path to liberation.

### 4.2.3. Tsunma Pema

Tsunma Pema's life story speaks specifically to the fragility of one's own vows and a *tsunma's* concern for practicing "*bodhicitta* (cultivating the mind of enlightenment for the benefit of others) as one cherishes one's own life."[29] Nearing the age of thirty, Tsunma Pema left her mother's house in Bhutan for Siliguri in northeastern India in the middle of night. She made her way by train to southern India where she entered a Nyingma monastery and received her vows from her root lama (Tib. *rtsa ba'i bla ma*), Penor Rinpoche. Eventually, due to health problems associated with the heat, she was moved to Himachal Pradesh to a Kagyu nunnery. The middle of three daughters, Tsunma Pema wanted to ordain for many years, but she waited until her own daughter was old enough to join a boarding school in

Sikkim before she departed for her monastic life. "I had her when I was twenty-three," she explains. While she recounts her mother's sadness and other relatives' surprise about her ordination, there are no undertones of regret when she retells her long-awaited desire to become ordained. Instead, she comments that since coming to India, she has finally had the opportunity to receive teachings from many lamas, "a sign of her great faith," her father says. "I had never heard of Mingyur Rinpoche before now. And now, he is teaching me meditation. And it's true what he says, you should do meditation whether you are walking, cooking, talking. Anything can be meditation."[30]

In a follow-up conversation, Tsunma Pema stated she wished she were a kind-hearted woman who would help all sentient beings in the world. She explained that previously she had "landed in the land of lay women" where she had fun, was crazy, and would "go out and screw up," however, "taking monastic vows helped to find the causes of her problems and learn from her mistakes."[31] Tsunma Pema also reported, "I am praying to the three Triple Gems not to depart from practice and my practice has to improve day by day in my life."[32]

Several months after our initial interviews, Tsunma Pema left her Kagyu monastery for reasons that are not entirely clear to me. She stayed with my family temporarily while a senior *tsunma* helped search for accommodations for Tsunma Pema. According to Tsunma Pema, someone accused her of indiscretions which were "all gossip and lies," she told me. This speaks to the fragility or perceived fragility of a *tsunma*'s vows. From my understanding, she was accused of having a relationship and this accusation alone resulted in her removal from the nunnery. Her vows, even if intact, were fragile-by-association.

After temporarily staying at another nunnery in Himachal Pradesh, she relocated to Bodh Gaya for her preliminary practices (*ngöndro*) including 100,000 physical prostrations while reciting refuge prayers and 100,000 mandala offerings for generating generosity.[33] She resided in Bodh Gaya from September 2018 until March 2020, practicing in the main temple when it opened at 5 a.m. until 11 a.m., followed by lunch and rest at her guesthouse, and then practice again from 3 p.m. until 7 p.m.

She explained in a phone call that due to COVID-19, Bhutan required all monastics who were Bhutanese nationals residing in India to return to Bhutan, but she intends to complete her final stages of *ngöndro* in Bodh Gaya when she is permitted to travel again. While she can practice at home, she is separate from not only the sacred site of the Buddha's enlightenment, but also being in the presence of another Nyingma teacher upon whose instruction she relies. While her family members practice Buddhism, the community of *ngöndro* practitioners at the temple also serve a family-like function where the reciprocity of merit-making appears bi-directional. For instance, at 6 a.m., rotating groups serve *balep* (Tibetan flat bread) and *daal* (Indian lentils mixed with spices) to practitioners while international laity from Nepal, Korea, Russia, and Vietnam walk around to chanting, meditating, or prostrating *tsunma*s and *tsunpa*s, offering candy, packs of cookies, fruit, and water. Those making offerings receive virtuous merit for their generous actions, and those practicing also generate merit by receiving these goods while simultaneously offering prayers and engaging in merit-making practices. Tsunma Pema often invited me to sit on her prostration platform while she took breaks after 1000 prostrations. She peeled her *mausami*, passing me slices of this sweet tangerine-lime like fruit while reminding me of ways I, too, as a lay person could generate merit, not only for my own benefit but also for others.

Presently, Tsunma Pema resides with her family in Bhutan where she spends time knitting intricate patterns and helping maintain the family garden.[34] However, she spends most of her day making Green Tara prayers to alleviate suffering caused by the pandemic. As she wrote on her Facebook page in August 2020, "Pray for my completion of merit and hardship. Purification and compassion for all sentient beings to be pure in mind all around the world."[35] In a way, the fragility of her own vows is tied directly to the responsibility she feels for others. She also posted a photo of herself with the caption, "Myself growing increasingly sad and impatient at the absurdity of this bullshitting samsara!"[36] Her forceful

indictment of the cyclical nature of rebirth speaks to her concern for the well-being of others and her own meditation on impermanence.

During Tsunma Pema's first interview, she told me we would talk about *gelongma* ordination later. She opted to skip the questions and suggested I interview other *tsunmas*. Tsunma Pema and I did not have conflicting interests—it genuinely seemed to be a topic that she chose not to speak on. We never discussed *gelongma* ordination again; Tsunma Pema, however, took on the role of recruiting other *tsunmas* to interview, often translating when I misunderstood or clarifying my English-accented Tibetan for *tsunmas*. She became more than a co-researcher; she is my friend.

I still do not seek her opinion on the topic of *gelongma* ordination even though I have had ample opportunities to do so—between drinking tea at her nunnery, taking her to a dentist, sharing fruit at the Mahabodhi temple, long walks to the commemorated site where the Buddha broke his fast when Sujātā offered him rice milk, and our frequent texts or phone calls every few months. More recently, she offered advice when my husband and daughters had COVID-19, insisting that my daughters attend school only online. She sends images of offerings on Guru Rinpoche Day or selfies while doing *khora* (circumambulating holy sites). Recently, we compared the January snowfall amounts in Thimphu and Chicago through photos and videos. She centers cultivating her practice and guarding the fragility of her own vows. Through our friendship and as I witness her practice, her silence on the topic of *gelongma* ordination makes sense to me.

## 5. When Institutional and Personal Fragility Collide

Tsunma Pema's silence on *gelongma* ordination fades into the background when observing her day-to-day practice. But how do I make sense of the Karmapa's silence on his postponed efforts to restore *gelongma* ordination? I understand the silence as suggestive of the fragility of the institution of Buddhist monasticism and one's own personal vows. As Ahmed says, "What happens to you: we need to handle what we come up against. But what if the handle is what breaks? Fragility: losing the handle. When the jug loses its handle, it becomes useless" (Ahmed 2017, p. 170). This is my projection, my near journaling, my moment of being Nao.

### 5.1. Institutional Fragility and Dōgen's Time-Being

*Gelongma* ordination is fragile because of its institutional history, a topic the Karmapa has addressed closely and carefully.[37] He ultimately decided that a dual-*Vinaya* approach with a fully ordained women's *saṃgha* would garner the most authentic vows. For instance, on 13 March 2017, during the Annual Nuns Gatheng, he stated,

> Without a *saṃgha* of *bhikṣunī*s, it is very difficult to give proper and authentic vows to individuals in a female body. This is why it is extremely important to reinstitute the community of *bhikṣunī*s. When we look at the ceremonies for the vows that can be taken with a female body—such as a female lay practitioner with precepts, the female who has gone forth, and the novice nun—as they are described in the *Vinaya* (and many of the ceremonies have been translated from Sanskrit into Tibetan), they all state that the *bhikṣunī*s, should bestow the vows that can be given to women. If we seek these authentic true vows, then the *bhikṣunī*s, are key.[38]

While the Karmapa pursued the possibility of restoring *gelongma* ordination through a dual-*Vinaya* approach with fully ordained Taiwanese *bhikṣunī*s, this process stalled without any discussion from 2017 until 2022.[39]

The Karmapa had difficulties returning to India after traveling to North America in summer 2017 to teach. During this time, he received a foreign passport without returning his Tibetan identity card, which created issues with receiving the necessary visa for re-entry into India. Due to these on-going challenges with the Indian government, the Karmapa was not able to return to his home monastery in India and was not able to attend in-person the annual Kagyu Monlam or the main gathering for *tsunmas* in 2018, 2019, or

2020 (Bagchi 2018; Karmapa 2019). To the best of my knowledge, he remains abroad. I could speculate that his inability to be physically present may be a possible reason that the *gelongma* ordination remains paused. The Karmapa has continued to teach through the COVID-19 pandemic and held the *tsunmas'* gathering online in 2021 and 2022. Therefore, even if the lineage of *gelongma*s "broke" when Buddhist monasticism began in Tibet, the Karmapa initially did not consider the fragility of *gelongma* ordination as preventing its restoration from happening.

Alternatively, does the stalling on this topic signify some sort of 'institutional inertia' on giving *gelongma* vows and restoring its lineage? Arguably, from the 8th century in Tibet until the 21st century in India, there has been a situation of, to borrow Ahmed's phrase, 'institutional inertia' (Ahmed 2017, p. 97). For roughly 1300 years, the *gelongma* lineage has remained non-existent for Tibetan and Himalayan *tsunmas*. While fragility may be one lens to think through such inertia, the notion of time and being offers a poetic, Zen-like lens for rendering understanding. Here, I think about the inseparability of time and being through the characters, Nao and Jiko. They rely on the 13th century Sōtō Zen monk, Eihei Dōgen who explores time and being in his *Shōbōgenzō Uji*:

An old Buddha said:

For the time being, I stand astride the highest mountain peaks.

For the time being, I move on the deepest depths of the ocean floor.

For the time being, I'm three heads and eight arms.

For the time being, I'm eight feet or sixteen feet.

For the time being, I'm a staff or a whisk.

For the time being, I'm Mr. Chang or Mr. Li.

For the time being, I'm the great earth and heavens above.

The "Time Being" means time, just as it is, is being, and being is all time.

The sixteen-foot golden Buddha-body is time; because it is time, it has time's glorious golden radiance. You must learn to see this glorious radiance in the twelve hours of your day. The [demonic asura with] three heads and eight arms in time; because it is time, it can be in no way different from the twelve hours of your day. Although you never measure the length or brevity of the twelve hours, their swiftness or slowness, you can still call them the twelve hours. As evidence of their going and coming is obvious, you do not come to doubt them. But even though you do not have doubts about them, that is not to say you know them. Since a sentient being's doubtings of the many and various things unknown to him are naturally vague and indefinite, the course his doubtings take will probably not bring them to coincide with this present doubt. Nonetheless, the doubts themselves are, after all, none other than time.

We set the self out in array and make the whole world. We must see all the various things of the whole world as so many times. These things do not get in each other's way. Because of this, there is an arising of the religious mind as the same time and it is the arising of time of the same mind. So it is with practice and attainment of the Way. We set our self in array, and we see that. Such is the fundamental reason of the Way–that our self is time (Roberts 2018, pp. 25–26).[40]

Dōgen's *Shōbōgenzō* emphasizes the present moment. This text reflects the Buddhadharma for Mahāyāna practitioners like Jiko who teaches Nao. As Nao writes, "Jiko also says that to do zazen is to enter time completely" (Ozeki 2013, p. 183). Dōgen teaches about the nature of delusions regarding time and being, especially when an ordinary person entrapped in dualistic and conceptual thought fails to understand that "the time-being abides in each moment."[41] Dōgen goes on,

Hence, pine trees are time. So are bamboos. You should not come to understand that time is only flying past. You should not only learn that flying past is the

virtue inherent in time. If time were to give itself to merely flying past, it would have to leave gaps. You fail to experience the passage of being-time and hear the utterance of its truth, because you learn only that time is something that goes past.

The essential point is: every being in the entire world is each time and [independent] time, even while it makes a continuous series. Inasmuch as they are being-time, they are my being-time (Roberts 2018, p. 27).

Dōgen teaches us to think about the present moment and not the past. This passage is meant to encourage the practitioner to respond in the moment to the needs of each situation removed and unfettered by our ideas and experience (Roberts 2018, p. 114). In this way, Dōgen's lens of the "time-being as moments" helps think through the fragility of *gelongma* ordination from the 8th century to the present day even though the contexts are different. The inertia remains because "all existence is included in each being-time" (Roberts 2018, p. 107). This means that if we know something about one particular time it facilitates recognizing the nature of time, which means "although we may get caught in our ideas about the difficulties, we are still being present as best we can" (Roberts 2018, p. 113).

### 5.2. Holding Contradictions

In the context of Ozeki's novel, Dōgen acts as an interlocutor for both Jiko and Nao. For purposes of this article, Dōgen's practices and own life narrative are instructive for holding contradictions. In Shinsu Robert's commentary on Dōgen's *Shōbōgenzō Uji*, she notes "We are so prone to understanding our experience as sequential that we have a difficult time including what we think of as contradictory ideas" (Roberts 2018, p. 114). Dōgen has his own paradoxical record with women, ranging from unconditional support of women to rejection (Arai 1999, pp. 22–23).

Paula Arai's instrumental work on Sōtō Zen nuns not only looks at Dōgen's views and support of women practitioners, but also illustrates how nuns interpret Dōgen's teachings. Dōgen's writings have perplexed scholars since some of his works suggested egalitarian ideals whereas others implied that women did not have the capacity for true enlightenment (Arai 1999, p. 39). Arai argues, however, that the historical record and nuns' present-day reliance upon his teachings indicate his past and present influence upon practitioners despite of the contradictions apparent in one sentence of the *Shukke Kudoku*. This sentence states, "'It is also said that one can attain Buddhahood in a female body, but this is not the Buddhist path of the true tradition of the Buddhist masters'" (Arai 1999, p. 39). Scholars question dating of this text and whether this sentence was edited because it is hard to hold contradictory views of an admirable teacher, scholar, and practitioner.

Even though Dōgen never ordained women, many women chose to transfer into his order because of his emphasis on the equality of female and male practitioners, and his successor (Keizan Jōkin Zenji) continued Dōgen's teachings on equality and ordained many women (Arai 1999, pp. 42–43). Arai and her nun interlocutors challenge interpretations that suggest Dōgen rejected women (Arai 1999, p. 49), but the paradoxes in Dōgen's own narratives and his actions help segue into thinking about the contradictions in the Karmapa's present situation. On the one hand, the Karmapa frequently speaks about gender equality. When we talk about equality of women, he says, it must go beyond addressing institutional inequality as it is not enough to really "empower women" (Karmapa 2015b, Wisdom Podcast). He has also noted that re-establishing *gelongma* ordination will not necessarily bring about gender parity in the same way that women's suffrage or electing a woman president do not truly restore women's rights (Karmapa 2015a, Princeton Talk). Thus, he ponders gender equality and its own paradoxes. While there was a prolonged silence about restoring the *gelongma* vows for five years, the Karmapa has not been silent on the fragility of vows. In 2019, he spoke about his own vows and the way he received them.

*5.3. The Form of Vows and the Karmapa on the Fragility of Vows*

For Tibetan Buddhist practitioners, there are three vows (*sdom pa gsum*)—*prātimokṣa* (*so thar gyi sdom pa*, individual liberation), *bodhisattva* (*byang sems kyi sdom pa*, adherent of enlightenment), and *tantra* (*gsang sngags kyi dam tshig*, esoteric continuum) (Wangyal 1996, pp. VIII–XI; Karthar 2004, pp. 151–80; Sobisch 2002). More importantly for the purposes of this discussion, the *prātimokṣa* vows are key for any vow holder. There are seven categories of vow holders to be distinguished by the person receiving them: fully ordained male monastic (*gelong/dge slong, bhikṣu*), fully ordained female monastic (*gelongma/dge slong ma, bhikṣūnī*), male novice (*getsul/dge tsul, śrāmaṇera*), female novice (*getsulma/dge tsul ma, śrāmaṇerikā*), layman practitioner (*genyen/dge bsnyen, upāsaka*), laywoman practitioner (*genyenma/ dge bsnyen ma, upāsikā*), and female monastic probationer (*gelobma/dge slob ma, śikṣāmanā*) (Kongtrul 2003, p. 88). In Tibetan epistemology, the *prātimokṣa* vows are included under the rubric of the form (*skandha*), albeit as subtle form, and *skandhas* by their nature are impermanent, changing moment to moment (Kongtrul 2003, pp. 85–87).[42] In this regard, the monastic tradition itself holds these *prātimokṣa* vows as fragile. As form, they are a part of our world, the desire realm, this *'jig rten*, this "disintegrating support."[43]

In *A Clear Differentiation of the Three Codes*, a translation of three-vow literature composed by Sakya Pandita in the thirteenth century (c. 1232), he draws from the *Abhidharmakośa* which delineates how the *prātimokṣa* vows are considered to be form (Sakya Pandita 2002, pp. 41–80).[44] The verses are as follows:

> From refuge through full monkhood,
>
> a Disciple's vows last as long as he lives.
>
> They are lost at death. (2)
>
> The effects of the vows
>
> manifest in a subsequent lifetime.
>
> The vows of a bodhisattva, however,
>
> endure even beyond death. (3)
>
> How so? A vow, Disciples maintain,
>
> is nonmental [i.e., material] and issues from body and voice;
>
> since it has form, the vow is relinquished whenever death occurs.
>
> On this point the *Abhidharmakośa* also teaches: (4)
>
> "The discipline of Individual Liberation is terminated
>
> by renouncing the training, by dying, by having become a hermaphrodite,
>
> by severance of the roots of virtue, and by the lapse of night."
>
> And this statement is authoritative (5).

Verses three and four emphasize the nature of the *prātimokṣa* vows in contrast with the *bodhisattva* vows, which endure beyond death. The *prātimokṣa* vows constitute a material, imperceptible form and do not carry over after death (Sakya Pandita 2002, p. 41).[45]

These conversations on the nature of vows also emerge in conversation with *tsunmas* and in the Karmapa's talks. Sitting in the shade in the back of the Kagyu Monlam pavilion in Bodhgaya, Tsunma Wangmo, a Taiwanese *tsunma* who practiced in the Tibetan tradition and was acting as a translator between Taiwanese, Himalayan, and Tibetan *tsunmas*, said:

> Vows are kind of like a *bum pa* (vase). During ordination, there is something kind of there that has a form, and it will come into this vessel. This is the moment you get the ordination; it means you have it. So, they want to make sure that you really have it exactly . . . All the conditions must be complete. And then, we have to make sure that once the time you get it, time will ripen, and you will remember the time you get it. It is related and everything comes together and the time ripens.[46]



Tsunma Wangmo speaks of time, form, and intention. Regarding time, she considers how the vows arise and the "time ripens." For me, her reflections call to mind Dōgen: "Because of this, there is an arising of the religious mind as the same time and it is the arising of time of the same mind. So it is with practice and attainment of the Way. We set our self in array, and we see that. Such is the fundamental reason of the Way—that our self is time" (Roberts 2018, p. 26; Dōgen 1994).

Tsunma Wangmo also speaks to the ideas of vows as form and intention—mirroring both the *Sautrāntika* and *Mādhyamika* perspectives of these individual liberation vows. The *prātimokṣa* vows yield a complete transformation in the mental continuum (i.e., a form that comes into the vessel) because they consist of the intention at the time when one renounces unwholesome deeds (i.e., all conditions must be complete for it to ripen) (Kongtrul 2003, pp. 85–87). While she does not speak to the fragility of the vows themselves, the Karmapa's own language about his vows implies their fragility. As he stated in his Kagyu Monlam address in 2019:

> In 2002, when I was 16, His Holiness granted me the vow of intermediate ordina­tion. And on the day when he did so, he gave me both the vow of intermediate ordination and getsul at that same time. Our request was only for the interme­diate ordination, but he gave me both ordinations. He must have had a special reason for doing so. Though at the time, my thought was to first receive the intermediate ordination and to later receive novice ordination from Situ Rinpoche and Gyaltsab Rinpoche, His Holiness gave me both. There was some talk within our lineage of the importance of my taking the vows according to our own tradi­tion and that it wouldn't be quite right to do otherwise. But at that time, to be honest, I hadn't studied the Vinaya much. In actuality, the vow of intermediate ordination is not the actual monastic ordination. It is really just permission to wear the robes, the symbol of religious ordination. One sets aside the clothes of a layperson and takes up the symbolic robes of ordination, but it is not actual ordination (Karmapa 2019).

He went on to add:

> After this, much time passed while I was wondering whether I should receive the novice vows according to our Kagyu tradition or not and what to do about full ordination. Further, I also became very busy with the work of Kagyu Monlam. As I studied the Vinaya and my understanding of it gradually increased, I felt like my former way of approaching vows was not quite correct. I thought my previous manner of taking them was not right, and that if I really wanted to receive the vows in a pure way, I should start again from the beginning. Especially, if one wants to receive the vows purely into one's being, one needs stable renunciation and wishing for emancipation in one's being. Without this, it would be difficult to keep the vows in a stable manner. These days, it is as if we were just following the custom of taking monks or nuns vows, but it's actually very rare that one thinks deeply about this and wishes, from the depths of one's being, to ordain. I think many people must be wondering and talking about why I have not taken full ordination by now. From my side, the main thing is that if renunciation and wishing for emancipation has not truly arisen, the novice and full monks' vows will not be based on this ethical conduct that longs for liberation, and it would be difficult for them to result in perfectly pure ethical conduct—though there must be some benefit in holding the vows anyway (Karmapa 2019).

The Karmapa does not speak to the form of the vows but speaks to the necessity of stable renunciation and a longing for liberation as the basis for keeping the vows "in a stable manner." Thus, one's vows are not only fragile but also, possibly, unstable. The Karmapa went on:

> It is difficult to have stable renunciation and a mind with the stable longing for liberation. And without these, it is difficult to hold the vows in a completely pure

way. So I am trying to develop stable renunciation within my being. I am trying to develop a certain degree of true renunciation—it's difficult to generate a really high level—but if I can develop a certain degree of renunciation, I feel that I will be able to receive the vows of individual liberation in a full and complete way. Then, at the time of death, if I can die with the support of ordination, I feel my mind would be at ease. This is the high hope that I hold for myself and the reason things have been as they are up to now (Karmapa 2019).

Here, the 17th Karmapa speaks of the value of holding monastic ordination vows as a source of ease at the time of death even though these vows as an imperceptible form disintegrate at the moment of death. During this particular talk, the 17th Karmapa implies multiple fragilities that extend beyond vows. The 17th Karmapa spoke of many issues from the difficulties acquiring the proper visa to return to India to the value of meeting with the other figure venerated as an incarnation of the 17th Karmapa, Thaye Dorje, whose recognition has been seen as the source of a potential rift in the Karma Kagyu lineage (Karmapa 2019).[47] As the 17th Karmapa Ogyen Trinley Dorje continued:

Generally, people say that I am Karmapa, a buddha, a bodhisattva. They say what they say, but when I look for myself, all I see is an ordinary being with afflictions and faults, not someone who is free of faults and endowed with all the qualities, as others think. In any case, my wish to benefit the teachings and benefit beings has never waned, and at the very least, I pray that I will be able to benefit the teachings and beings not only in this lifetime but in all my lifetimes (Karmapa 2019).

He went on:

For a long time, no one has heard much about what I am doing, and there are many rumors and a lot of hearsay about it. We are all the same. No matter who we are, people say all sorts of things about us; they misunderstand or make up things. In my life, this has often happened from the time I was little until now. Such situations happen often to all of us. But the main thing is that because our own minds are not hidden from us, it is important for us to believe in ourselves. For me, as I said before, I intend to continue working for the sake of Buddhism and sentient beings (Karmapa 2019).

The Karmapa's address speaks to the form of vows and the fragilities that form entails. Are these fragilities like the cracks that let the light in?

*5.4. Personal Fragility and Dōgen on a Critical Instant*

*Prātimkoṣa* vows as form are personally fragile for Buddhist practitioners. More than that as Tsunma Wangmo and the 17th Karmapa both illustrated, the fragility of one's personal vows is also contingent on personal intention and the stability of the mind of renunciation. Dōgen and Nao, again, are helpful to think alongside while pondering personal fragility. Dōgen plays into the narratives Nao writes in her journal. For instance, Nao translates her great uncle's secret French diary from World War Two. Haruki #1, her great uncle, who died in a *kamikaze* mission, writes,

Dōgen also wrote that a single moment is all we need to establish our human will and attain truth . . . . Both life and death manifest in every moment of existence. Our human body appears and disappears moment by moment, without cease, and this ceaseless arising and passing away is what we experience as time and being. They are not separate. They are one thing, and in even a fraction of a second, we have the opportunity to choose, and to turn the course of our action either toward the attainment of truth or away from it. Each instant is utterly critical to the whole world (Ozeki 2013, p. 324).

Dōgen is helpful to think with here, especially considering not only the links between moments and their perpetual impermanence but also how "each instant is utterly critical." Haruki #1's interpretation of Dōgen signals to the importance of each moment and the

inseparability between time and being and action. Thus, through Dōgen read through Ozeki's characters, what is the course of action when considering the fragility of vows and *gelongma* ordination?

## 6. Super Nuns for the Time Being

Nao's journal tells the life story of her great-grandmother Jiko—a nun, novelist, New Woman of the Taisho era, an anarchist, and a feminist (Ozeki 2013, p. 6). Nao's insights about Jiko's outlook may bring some resolve for the time being:

> I believe that in the deepest places in their hearts, people are violent and take pleasure in hurting each other. Old Jiko and I disagree on this point. She says that according to Buddhist philosophy, my point of view is a delusion and that our original nature is to be kind and good, but honestly I think she's way too optimistic. I happen to know some people, like Reiko, are truly evil, and many of the Great Minds of Western Philosophy back me up on this. But still I'm glad old Jiko believes we're basically good, because it gives me hope, even if I can't believe it myself. Maybe someday I will (Ozeki 2013, p. 181).

Jiko gives Nao hope.

The *tsunmas* at the center of this case study offer a similar hope for other *tsunmas* and the community of lay women they support. Tsunma Nyima Drolkar, the abbess of a Geluk nunnery in Himachal Pradesh, explains her intentions and motivations as a *tsunma*:

> My motivations are for making my monastery a simple monastery because I do not want [the other *tsunmas*] to live as I did before. In the beginning, we woke up at five o'clock in the morning and worked the whole day like laborers. So, we worked with sand and bricks which we all carried on our backs. Now, I wish for the other *tsunmas* to just study. My intentions and motivations are that everybody feels this monastery models good practice and study. That is my dream.[48]

Tsunma Nyima literally and figuratively lays the foundation for the other *tsunmas* in her community. Her motivations draw from her wish to work on behalf of the other *tsunmas*. When I asked, "How do you see yourself and your role in your monastic community?" She noted, "Actually, I am abbess for the nunnery. I am very happy with this role as I like to help others. The young *tsunmas* call me their second mother."[49]

When asked about her responsibilities as a *tsunma*, Geshema Jigme said,

> First our responsibility is to practice the Buddhadharma very deeply much like Jetsun Milarepa. Secondly, we teach *tsunmas* or lay women. I think like that. *Tsunpas* do not need our help since there are so many *geshes* and *khenpos*. We are all the same. All women. That's why we help them. We have more time because we have a simple life. We don't look after parents. Lay people have so many problems. They have to look after their children, spouse, and parents. That's why we teach lay women how to practice Dharma and keep good health.[50]

Broadly, her response reflects her commitments to helping women in her personal, interactional world. She generalizes about "women" whom she sees as similar in their "womanness." Geshema Jigme intends to support the differing needs of monastic and lay women. Her care for them is similar to Jiko's care for Nao. Nao writes:

> By the end of the summer, with Jiko's help, I was getting stronger. Not just strong in my body, but strong in my mind. In my mind, I was becoming a superhero, like Jubei-chan, the Samurai Girl, only I was Nattchan, the Super Nun, with abilities bestowed upon me by Lord Buddha that included battling the waves, even if I always lost, and being able to withstand astonishing amounts of pain and hardship (Ozeki 2013, p. 204).

Nao makes an important point. She, too, lives with fragility—astonishing amounts of pain and hardship, but she is a "Super Nun." In spite of the consequences of Nao's fragility and the fragility of the *gelongma* lineage in Tibetan history, there are plenty of Super Nuns!

Again, Ahmed's position on fragility provides a useful lens:

> Perhaps we need to develop a different orientation to breaking. We can value what is deemed broken; we can appreciate those bodies, those things, that are deemed to have bits and pieces missing. Breaking need not be understood only as the loss of integrity of something, but as the acquisition of something else, whatever else that might be (p. 180).

The *gelongma* ordination lineage is fragile, but we can think about what ordination possibilities may come in due time. Clearly some of those possibilities have come to fruition through the efforts of the Je Khenpo of Bhutan and the more than one-hundred and forty *gelongma*s who received their vows through a single *saṃgha* ordination. Additionally, during the 8th Arya Kshema in April 2022, the 17th Karmapa spoke to his interest in continuing with the process towards *gelongma* ordination (Karmapa 2022).[51] As my interlocutors own daily practices demonstrate, Himalayan and Tibetan *tsunmas* have been *tsunmas* for 1300 years. Whether a *gelongma* lineage emerges from a *gelong*-only or dual-*Vinaya* ceremony remains unclear. The *tsunmas* at the center of this case study have developed different orientations towards the breaking, the fragility of the lineage, but as Geshema Jigme says:

> We Tibetans accept *bhikṣuṇī*s from Korea, Hong Kong, Taiwan, and the west. We accept all of them. We are also like them, but why do we not accept our own *tsunmas* as *bhikṣuṇī*s (*gelongma*s)? If Tibetan *tsunmas* become *bhikṣuṇī*s (*gelongma*s) and there are no discussions on it, then that is not a good sign. If we have discussions on it, it shows we are actually focusing on them. If my pen is broken, the majority of people think it is not good because my pen is broken. But I do not think like that. I think that since it is broken, it is a good chance to get a new one! I think like that always![52]

Maybe it's time for new pen, a new *gelongma* lineage. Or, to return to the fire metaphor that I began with—maybe it's time for building fires in new ways. Maybe those fires are being lit by the Je Khenpo of Bhutan. If you prefer to end in song, "Ring the bells that still can ring, Forget your perfect offering, There is a crack, a crack in everything, that's how the light gets in."

**Funding:** This research was funded by Fulbright-Nehru Grant 2017–2018.

**Institutional Review Board Statement:** The study was conducted in accordance with the Declaration of Helsinki, and approved by the Northwestern University Institutional Review Board (protocol code STU00205244 and date of approval: 13 September 2017).

**Informed Consent Statement:** Informed consent was obtained from all informants involved in the study. Per guidelines of the IRB at Northwestern University, all informants names have been changed for confidentiality and to maintain anonymity.

**Acknowledgments:** The author appreciates the insightful feedback from presenting earlier versions of this project initially at Northwestern University's Department of Religious Studies Works- in-Progress Workshop organized by Michelle Molina as well as the support of Sarah Jacoby and Rob Linrothe during the Asian Studies Graduate Cluster Speaker Series. I am especially grateful for the conversations and suggestions offered during the Gender Asymmetry in the Different Buddhist Traditions Conference at the University of Perugia, Italy organized by Ester Bianchi and Nicola Schneider. I would also like to thank D. Mitra Barua, Annie Heckman, and Karma Lekshe Tsomo for insightful comments on earlier versions of this essay.

**Conflicts of Interest:** The author declares no conflict of interest.

## Notes

1    For more on this topic of the first ordination and the potential for decline in different *vinayas*, see Anālayo (2010, pp. 78–97).

2    See other forthcoming articles in this Special Issue for more regional specific historical information: for *Theravāda/Pāli*, see Bhikkhuni Dhammadinnā, Scott, Seeger, and Walker; for *Dharmaguptaka*, see Campo, Cho, Bianchi, and Pérronet; for *Mūlasarvāstivāda*, see Bareja-Starzynska, Schneider, and Wu.

3   For more on the history of *Theravāda/Pāli*, see Seeger and Bhikkhuni Dhammadinnā in this Special Issue. Also, see works by Bhikkhunī Kūsuma (2012) and Bhikkhunī Dhammandandā (Kabilsingh 1991), who are two *Theravāda bhikkhunī*s who also received their full vows with *Dharmaguptaka* monks. Other additional scholarly works that examine full ordination in the *Theravāda/Pāli* context include (Collins and McDaniel 2010; Battaglia 2015; Kawanami 2007; Mrozik 2009, 2014, 2020; Salgado 2013, 2019; Seeger 2018).

4   I use the term *tsunma* because it covers all celibate women religious practitioners who hold precepts in the Tibetan tradition, whether these women have postulant (*rab tu byung ma*, Skt. *pravrajitā*), novice (*dge tshul ma*, Skt. *śrāmaṇerikā*), probationary (*dge slob ma*, Skt. *śikṣamāṇā*), or full vows (*dge slong ma*, *bhikṣunī*), and whether or not they don monastic attire, shave their heads, and live in or apart from a monastic community. Yet all these Tibetan terms—*tsunma, ani, jomo, rabjungma, getsulma, gelobma*, and *gelongma*—are often considered equivalent and interchangeable with the English term nun. Tsunma Tenzin, an interlocutor in Bodh Gaya, used the term *tsunma* in lieu of the more common *ani*. The Tibetan term *btsun* is equivalent to discipline. As Tsunma Tenzin discussed, *btsun ma* refers to women who adhere to discipline. Interview 3F–TT–Bihar—16 March 2018, 20:49–22:44. All interlocutors names are pseudonyms. I do not use the names and exact locations of nunneries for purposes of confidentiality under the guidelines of the Institutional Review Board at Northwestern University. I conducted all interviews on my own in Tibetan, Hindi, and/or English. Dorjee Choephel was instrumental in transcribing interviews, which we subsequently edited together.

5   The 17th Karmapa Ogyen Trinley Dorje initiated this process on 11–12 March 2017, fieldnotes. Presently, the 17th Karmapa has a court case for spousal support against him amid allegations of sexual assault that emerged in May 2021. According to a *CBC News* article that first published these allegations, Vikki Hui Xin Han is suing for spousal support based upon what "began as a non-consensual sexual encounter [that] evolved into a loving and affectionate relationship" (Proctor 2021). Initially, Han sued for child support but amended her claim for spousal support when the case went to trial (Müller 2021; Burke 2021). Since these allegations postdated all of my fieldwork, I do not know how the *tsunmas* at the center of this study have received this knowledge of the allegations against the Karmapa.

6   I am using this term vow as it relates to the acceptance of the pledge to not break the precepts (Tib. *bslab pa'i gzhi*, Skt. *śikṣāpada*) that *tsunma*s practice. There are complex linguistic issues on this topic of the Tibetan term *sdom pa*. For more discussion, see Kishino (2004). See also Roloff's (2020) discussion, pp. 281–93, and footnote 71 on p. 308. In terms of the enumeration of the precepts (Tib. *bslab pa'i gzhi*, Skt. *śikṣāpada*), this is by no means fixed, as in Tibetan exegesis or as illustrated in recent scholarship. For more on this topic, see Heckman (2022, chp. 2–3).

7   Tsering's translation is presumably referring to the Tibetan term, *dag pa*, as does Clarke's (2010, p. 235). Bhikṣunī Tsedroen and Bhikkhu Anālayo also suggest that the preceptor giving the vows does not incur a minor fault (*nyes byas*) (Tsedroen and Anālayo 2013, p. 761). The point is that since the preceptor incurs an infraction, the vows are not faultless and perfect. However, the ordinations are still valid. Clarke reiterates this point that there are canonical instances where even slight mishaps or even non-ideal ordinations are still valid (Clarke 2010, pp. 232, 237–38). Thanks to Annie Heckman for thinking this through with me. See also Salgado (2019, pp. 4–8). Salgado discusses the nuances of the precepts and the cultivation of self-discipline by showing the limitations of characterizing *Vinaya* stipulations as rules. For discussions on this topic in a Mahāyāna context, see Péronnet (2022).

8   See Roloff's (2020) text as a reference for the English translation of the ordination procedures for *bhikṣunī*s from the *Kṣudrakavastu*.

9   With gratitude for Professor Richard Kieckhefer making this connection for me during our "works-in-progress" workshop at Northwestern 2/22/22. Leonard Cohen, *Anthem*, 1992 (Cohen 2022).

10  Though there are other sources, here, I am relying on the 8th Karmapa Mikyö Dorje's commentary (Karmapa 2002) on the *Mūlasarvāstivāda Vinaya*, the center of my current research. The full passage is as follows: "In the main regions, the female *saṃgha* requires twelve fully ordained nuns. In the borderlands (remote area), in the case that twelve nuns are not obtained, then six fully ordained nuns are permissible. Even if these numbers are not complete, given with a group of four fully ordained nuns and without making a mistake in the confession in the monastic discipline is to know the essential point described in the full ordination. If these fully ordained nuns are not obtained in the region, it is suitable for the *saṃgha* of fully ordained monks to also bestow the full ordination to women." dge 'dun ma de yang yul dbus su dge slong ma bcu gnyis dang ǀ mtha' 'khob tu bcu gnyis med na dge slong ma drug gi grangs tshang ba'o ǀ grangs de dge ma tshang yang dge slong ma bzhi'i tshogs kyis byin na tshangs spyod nyer gnas 'chogs la nyes byes te ǀ bsnyen rdzogs la gsungs pa'i gnad kyis shes so ǀ dge slong ma de dge ma rnyed na ǀ dge slong pha'i dge 'dun kyis kyang tshangs spyod nyer gnas sbyon du rung ste ǀ.

11  Tibetan histories detail how the *gelong saṃgha* required reinstatement following Lang Darma's reign, insinuating how the *gelong* ordination lineage is also fragile. Yet, a *gelongma* ordination is a multi-year process with several stages—preventing it from happening from the 8th century to the present moment whereas a *gelong* ordination does not have a probationary stage. Thus, *Vinaya* requirements for the male versus female *saṃgha* create the conditions for differing degrees of fragility between the *gelong* and *gelongma* ordination.

12  Generally, Mahāyāna practitioners in China, Hong Kong, Japan, Korea, Vietnam, and Taiwan rely on the *Dharmaguptaka Vinaya* whereas Vajrayāna monastics in Bhutan, India, Mongolia, Nepal, and Tibet rely on the *Mūlasarvāstivāda Vinaya*.

13  For the perspective on "purity" in the *Dharmaguptaka* lineage, see Bianchi (2022).

[14] During the 8th Arya Kshema on 19 April 2022, the 17th Karmapa reasserted his wish to continue this *gelongma* ordination in the future when it was safe after the COVID-19 pandemic.

[15] I closely detail several exceptional cases of *gelongma*s in Tibetan history in Chapter 3 of in my dissertation, illustrating how some *tsunma*s speak of these uncommon *gelongma* narratives as possible precedents for restoring ordination.

[16] Interview 22F–LWD–UT—28 July 2018, 1:04–1:07.

[17] Interview 22F–LWD–UT—28 July 2018.

[18] Interview 3F–TT–Bihar—16 March 2018, 6:00–9:00.

[19] Interview 3F–TT–Bihar—16 March 2018, 40:00–44:00.

[20] Interview 3F–TT–Bihar—16 March 2018, 46:00–48:00.

[21] As I discuss in Chapter Four of my dissertation, Löpon Wangmo would like to receive her *gelongma* vows but does not see a viable option. In contrast, Tsunma Tenzin would like to keep perfecting her novice vows and did not express an interest in receiving full vows.

[22] Gelongma Sonam Khacho's life story and full ordination are central to my dissertation (Price-Wallace 2022).

[23] Interview 8F–GS with TP–HP—24 April 2018, 54:30–58:50.

[24] Interview 1F–GS–HP—22 January 2018, 10:40–12:50.

[25] Hanna Havnevik's (1989) important ethnography, *Tibetan Buddhist Nuns: History, Cultural Norms and Social Reality*, details many important aspects of these *tsunma*s' lives, pp. 90–113.

[26] Interview 1F–GS–HP—22 January 2018, 34:41–32.25.

[27] Interview 25F–TY–UT—30 July 2018, 18:17–23:50.

[28] Interview 25F–TY–UT—30 July 2018, 29:35–31:40.

[29] Survey with Pema, 2C, 2 March 2018–Bihar.

[30] Interview 8F–TP, informal 22 February 2018, 7 March 2018–Bihar, 24 April 2018–HP, 20 October 2019–Bihar, no recording.

[31] Survey with Pema, 2C, 2 March 2018–Bihar. She wrote these answers in English.

[32] See note 31 above.

[33] *Ngöndro*, meaning 'that which comes first,' is the foundational practice containing essential Buddhist teachings. *Ngöndro* is an essential practice among all the Tibetan Buddhist schools although each has their own variations of the ordinary preliminaries such as the four thoughts that turn the mind and unique preliminaries such as refuge, purification, accumulating merit, and guru yoga. For more on this topic, see Patrul Rinpoche (1994) and Mingyur Rinpoche (2014).

[34] Tsunma Pema, WeChat messages, 2020.

[35] Tsunma Pema, Facebook post, 21 August 2020. All posts are recorded as she wrote them.

[36] See note 35 above.

[37] The Karmapa has addressed this topic thoroughly during the 1st, 4th, and 8th Arya Kshema gatherings.

[38] The 17th Karmapa, Arya Kshema, Bodh Gaya, India, 15 March 2017, translation by Michele Martin. Field notes, 15 March 2017.

[39] He has been exploring this topic since 2015 and made a formal announcement during the Arya Kshema Gathering. For more on this topic, see Roloff (2020, p. 329). See also Bianchi (2022).

[40] Roberts draws from Norman Waddell and Masso Abe's translation (Waddell and Abe 2002). For alternative translation, see Dōgen (1994).

[41] Welch and Tanahasi's translation.

[42] Different philosophical schools assert various positions as to how this "form" is understood. There is also a debate on this topic by Chinese Vinaya scholars. See Newhall (2014).

[43] With gratitude to an anonymous reader for her clarifying points on this topic.

[44] Sakya Pandita draws directly from the *Abhidharmakośa*. His stance accords within the *Mūlasarvāstivāda* perspective of the Śrāvaka *prātimokṣa* since it is the only one that existed among Tibetan monastics. It is also what he received from his abbot, Śakyaśrībhadra (Sakya Pandita 2002, p. 22). See also note on page 73 of the book.

[45] Also see Khenpo Khartar Rinpoche on Karma Chakme, who also delineates how the different schools such as Geluk and Kagyu differ in their understanding of how and when the three vows are received.

[46] Collaborative interview completed with Michele Martin. We interviewed Tsunma Wangmo, 6 March 2017, Bodh Gaya, India. Interlocutor's name has been changed for confidentiality.

[47] Both wrote letters signaling to aspirations to cease rifts among the Karma Kagyu lineage. For more, see Lefferts (2018).

[48] Interview 4F–TND–Bihar—18 March 2018, 12:20–18:24.

[49] See note 48 above.

[50] Interview 15F–GJ–HP—21 July 2018, 20:10.

51 For a nearly complete transcript, see the detailed write up from the publicity team of the 8th Arya Kshema who writes, "His Holiness expressed that in the future, when the epidemic has ended and we can once again travel easily, he would like to invite the bhikshuni sangha from another country again to give the novices the nun-in-training vows and then later the bhikshuni ordination. Within the practice lineage of Karma Kamtsang, this topic of bhikshuni ordination was not something he had decided alone, Karmapa clarified. It was a result of several conferences held during the Kagyu Gunchö among the khenpos, geshes, and students. At that time, the khenpos and geshes told him, "You should institute bhikshuni ordination in the Kamtsang Kagyu," and he heeded the requests." (Arya Kshema: Winter Dharma Gathering for Kagyu Nuns 2022).

52 Interview 15F–GJ–HP—21 July 2018, 1:59–2:02.

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
