# Peer review of "The Fragility of Restoring Full Ordination for Tibetan Tsunmas (Nuns)"

_religions, doi:10.3390/rel13100877_

Round 1
Reviewer 1 Report
This essay, based on interviews with Tibetan and Himalayan tsunmas and the 17th Karmapa, the religious head of the Karma Kagyu school of Tibetan Buddhism, aims to discuss perspectives on the controversial and important higher ordination of Tibetan Buddhist nuns and how it might be given to tsunmas. Among other concepts, the paper argues that Sara Ahmed’s concept of “fragility” can be used to help think about different monastic perspectives on the gelongma ordination, including its prohibition as well as its possibilities.
The use of the concept of fragility should be better related to its broader theoretical framework. A clearer cultural translation of the concept vis-à-vis perspectives on the gelongma ordination is necessary. If everything is fragile, as stated in lines 140-141, how is fragility even useful as an interpretive construct for the topic of the paper, (or any topic for that matter)?
The author needs to articulate if and how what the nuns say makes sense in relation to the broader context of Ahmed’s work. How is that context meaningful to the potential Vinaya practice of tsunmas and to their current practice? The concept of fragility should be contextualized in relation to other key ideas of relevance used by Sara Ahmed such as the “brick wall,” “complaint” and “being included” and then made relevant to the ordination issues that the author discusses. How does Ahmed’s ideas about the transformation of institutions in general relate to the transformation of monastic institutions?
Clarify the significance of the paper: Why does the question of higher ordination matter to Tibetan and Himalayan tsunmas? How might the gelongma ordination impact the actual practice of tsunmas? Since tsunmas now have access to the geshema exams, the author might expand on how that access influences views on the gelongma ordination and the current education of nuns.
Provide more context about the debate on the higher ordination among Tibetan traditions. The author draws from discussions with tsunmas from three different lineages: the Geluk, Kagyu and Sakya lineages, but has not indicated if there are distinctions in perspectives on the gelongma ordination arising from within these traditions and that might be unique to them.
The paper should incorporate a discussion of relevant works presented at the pivotal 2007 “International Congress on Buddhist Women’s Role in the Sangha” held in Hamburg, and which focused on the higher ordination of Tibetan nuns. Additional works that should be incorporated include the seminal publications by Kim Gutschow and Hanna Havenvik.
Reference to discussion about the possibility of a dual ordination for tsunmas is indicative of similarities to discussions about the dual ordination for Theravāda Buddhist nuns. The author should engage scholarly publications on the recent establishment of Theravāda ordinations, since those ordinations were initiated among debates analogous to those relating to the gelongma ordination.
Sexual violence, as mentioned by the author, is a transformative experience documented in Ruth Ozeki’s novel. However, it has yet to be made relevant to the concerns of the tsunmas. Ozeki’s novel and Dogen’s thinking seem extraneous to the experience of the nuns who are interviewed and should either be more clearly integrated into the theoretical context of the paper and the practice of tsunmas or removed.
Author Response
Per the reviewer's suggestion, "The use of the concept of fragility should be better related to its broader theoretical framework," I brought in more of Ahmed's theoretical framework and underscored its importance as a tool for thinking about this topic.
The reviewer asked, "how is fragility even useful as an interpretive construct for the topic of the paper, (or any topic for that matter)?" I addressed this by adding, "Fragility has many nuances which I detail throughout article. Fragility is a useful interpretive construct for thinking about gelongma ordination since the breakage of this Mūlasarvāstivāda ordination lineage for women exhibits a “shattering of possibility.” This means that its breakability is reflective of its fragility and “what breaks off is on the way to becoming something” (2017 168, 186). Thus, the question of restoring gelongma ordination points to breakability as well as how fragility points to the possibility of something else. "
I address the other questions posed by reviewer about the significance of full ordination the new section at the beginning, which gives more history and background. I do not go into geshema exams as Nicola Schneider (the editor for this special edition) or Theravada ordination histories as Bhikkuni Dhamadinna and Martin Seeger are writing specifically about this in this edition.
The reviewer comments that, "Sexual violence, as mentioned by the author, is a transformative experience documented in Ruth Ozeki’s novel. However, it has yet to be made relevant to the concerns of the tsunmas. Ozeki’s novel and Dogen’s thinking seem extraneous to the experience of the nuns who are interviewed and should either be more clearly integrated into the theoretical context of the paper and the practice of tsunmas or removed." I in the significance of the use of literature with my new paragraphs in the beginning.
Reviewer 2 Report
A very promising contribution to the literature with demonstrably strong ethnographic skill and eloquent writing. However, the piece would greatly benefit from targeted revisions to strengthen the content presented, particularly a reorder to clarify foundational content early in the piece so the theoretical analysis and literary analogies can highlight the central argument and narrative arc. It would greatly benefit the piece to begin by presenting the history of ordination, particularly as it relates to the Mūlasarvāstivāda Vinaya and Dharmaguptaka Vinaya, with the specific regional and school-based nuances, alongside the details of the 7 different classes of vows. Then introduce the multi-year process and probationary period that applies to the Tibet context and how it prevented the establishment of gelongma ordination. Then provide an analysis of the arguments for and against instating full ordination given the three approaches presented (Mūlasarvāstivāda Vinaya only; ecumenical with fully-ordained monastic women under the Dharmaguptaka Vinaya; and the dual approach). After these foundations are set, Ahmed’s notion of fragility — “when being breakable stops something from happening” — will well-characterize the content of the article. However, currently, the reader experiences the notion of fragility being declared but not demonstrated with substantial content (i.e., the first half of the article reads as more telling than showing…).
Truly beautiful prose and well-articulated sections throughout the work. It seems this piece was extracted from a larger work (i.e., the dissertation) that provided the bigger picture. Yet this work still needs the larger narrative arc to provide foundations for the specific claims being made.
In addition to the framing of the notion of fragility and its relationship to key issues in the gelongma context, the hermeneutic of textual communities also needs further development. The initial framing with the Anne Blackburn quote starts from a confusing association where tsunmas are not allowed to study the Vinaya but are then characterized as a textual community in relation to the texts that shape their understandings of monastic discipline and identity.
The section on Dōgen’s Time-Being is poetic and likely frames the content nicely in the larger dissertation work. However, in the current framing, it reads a bit tangential to the central thread, as does the Nao interlude. I imagine they interweave beautifully in the larger dissertation. However, in this shorter article format, we are sacrificing much core content framing full ordination and its relationship to the notion of fragility for what reads as literary associations.
A reworking of the order of content and honing the key components of the gelongma context that contribute to the notion of fragility and this manuscript will be ready for publication. Attached are notes on the actual manuscript as well as minor edits throughout (e.g., insert years for in-text citations, remove numbers in final bibliographic list, check Wylie sections for minor errors, etc.).

Author Response
Thank you for your review. Based on Reviewer 2's comment, "A reworking of the order of content and honing the key components of the gelongma context that contribute to the notion of fragility and this manuscript will be ready for publication," I added a new introductory section that aimed to address both Reviewer 1 and 2's request for more contextualization and a history of ordination. I also framed why I was using the idea of fragility. In this regard, I also clarified why I find Ahmed helpful to think with, especially how she, too, draws from literary associations.
I also tried to clarify my language around Blackburn's use of textual communities. As Blackburn's work indicates, textual communities can also be non-literate. While my interlocutors are literate, they rely on the interpretative practices of their teachers and oral transmission rather than direct textual study.
I understand that the literary associations may be poetic, but I hope that this work can be useful for anyone interested in the intersections of religion and literature as well. Thank you for your time and your line edits in the text. I really appreciate the detailed read.
Round 2
Reviewer 1 Report
The improvement in this paper is significant and the introduction is better than before. This version also shows more in-depth use of the work of Ahmed. The article could yet be enhanced with attention to the following mentioned earlier:
Additional works that should be incorporated include the seminal publications by Kim Gutschow and Hanna Havenvik.
More substantial attention to scholarly publications on the recent establishment of Theravāda ordinations, would be useful. I recommend integrating discussions found in Nirmala S Salgado's "Buddhist Nuns and Gendered Practice."
There are minor spelling/typographical errors that need correction in this version.
Author Response
Havnevik, Gutschow, and Salgado have been and continue to be extremely influential for all of my work. All three shaped and inspired my dissertation and for that reason they are heavily cited in it and another forthcoming paper. I did not want to repeat myself in this publication but will draw attention to their works in my revisions.